# Continuous remote monitoring of postoperative vital signs with the Biobeat patch: A multicenter prospective observational study

Inès Kouidri[1], Cécile Landais[2], Audrey Solis[3], Delphine Petit[4], Aurélie Blondeau-Martin[1], Karine Merlin[1], Bernard Trillat[5], Morgan Le Guen [1,6], Marc Fischler [1]*

1 Department of Anesthesiology and Pain Medicine, Hôpital Foch, Suresnes, France, 2 Department of Epidemiology - Data - Biostatistics, Delegation of Clinical Research and Innovation, Hôpital Foch, Suresnes, France, 3 Department of Anesthesiology, Groupe Hospitalier Diaconesses Croix Saint Simon, Paris, France, 4 Department of Anesthesiology, Groupe Hospitalier Paris Saint-Joseph, Paris, France, 5 Department of Information Systems, Hôpital Foch, Suresnes, France, 6 Université Versailles-Saint-Quentin-en-Yvelines, Versailles, France

* m.fischler@hopital-foch.com

## Abstract

### Objective

This study, conducted under standard clinical conditions, evaluated the Biobeat wearable sensor in postoperative patients monitored over multiple days without human intervention. The study aimed to assess technical performance (vital sign data collection and artifact frequency), reliability (agreement with nurse-measured values), the sensor's ability to detect vital sign abnormalities, patient satisfaction, and incidence of adverse skin reactions.

### Design

This multicenter prospective observational study included patients who underwent abdominal surgery with a planned postoperative ward stay between December 2020 and December 2022.

### Methods

Alongside routine care–comprising (2–3 times daily nurse assessments of blood pressure (BP), pulse rate (PR), respiratory rate (RR), peripheral oxygen saturation (SpO2), and temperature)–participants wore a precordial Biobeat sensor. The sensor transmitted data wirelessly to a cloud-based repository via Wi-Fi without human intervention.

### Results

Of 109 enrolled patients, 90 were included in the analysis. The median recording duration was 63.8 hours (interquartile range: 42.6–72.3 hours). Data loss occurred in

**Data availability statement:** The data are available at Dryad (https://doi.org/10.5061/dryad.69p8cz9dh).

**Funding:** The author(s) received no specific funding for this work.

**Competing interests:** The authors have declared that no competing interests exist.

45% of the recording time, and pulse oximetry data were absent in 16% of the available measurements. Artifacts were infrequent, comprising 277 of 35,998 measurement points. Compared with nurse measurements, Biobeat showed small mean differences for most variables, except for RR and temperature, yet exhibited wide limits of agreement across all variables. Clarke Error Grid analysis revealed excellent concordance between Biobeat and nurse measurements for SpO2 and PR, with lower agreement for mean BP, RR, and temperature. The sensor's continuous monitoring detected more vital sign abnormalities, including severe hypotension (mean BP<60 mmHg), in 10.0% of patients compared to 3.3% with nurse measurements (P=0.04). The patients reported high satisfaction and no adverse skin reactions were observed.

## Conclusions

Significant data loss presents a substantial challenge for analysis, underscoring the critical need for improved data transmission and flow control measures in study protocols and clinical deployment.

## Registry

ClinicalTrials.gov (NCT04585178, October 14, 2020).

## Introduction

Recent advances in surgical care–including enhanced preoperative optimization, minimally invasive techniques, and streamlined postoperative management–have reduced hospital stays and expanded ambulatory surgery, even for complex procedures. However, postoperative complications remain a challenge, particularly as the patient population ages and presents with increasing comorbidities. Complication rates in general surgery range from 5.8% to 43.5%, with a 2018 study reporting an incidence of 12.5%, of which 60.3% were classified as severe (Clavien-Dindo grade ≥IIIA) [1]. The European Surgical Outcomes Study found that 73% of postoperative deaths occurred in patients who were never admitted to intensive care facilities, thereby underscoring the limitations of available resources [2]. Conventionally, vital signs are assessed every 8–12 hours on general surgical wards following a brief post-anesthesia care unit (PACU) stay, a practice that risks delayed detection of clinical deterioration [3]. Michard and Sessler described a continuum of postoperative monitoring, extending from intermittent ward checks to wearable sensors that enable practical, real-time tracking outside critical care settings. They cautioned against alarm fatigue from excessive false alerts and suggested that machine learning approaches could refine measurement accuracy [4].

Since the early 2000s, advances in broadband Internet, Wi-Fi, and Bluetooth technology have transformed remote patient monitoring from occasional telemedicine experiments into structured systems capable of continuously measuring and wirelessly transmitting patients' clinical parameters including blood pressure (BP), heart rate (HR),

pulse rate (PR), respiratory rate (RR), peripheral oxygen saturation (SpO2), and temperature. Numerous wearable devices can capture multiple vital signs simultaneously and transmit data in real time to monitoring platforms or electronic medical records. In a recent narrative review, Bignami and colleagues identified nine clinically validated multiparametric monitoring systems. Although distinctions are not absolute, several devices are designed primarily for in-hospital use (CheckPoint Cardio's CPC12S, Lifetouch, Portrait Mobile, Radius VSM, SensiumVitals, and VitalPatch [5–9]), whereas others are oriented toward out-of-hospital applications, particularly prehospital and home monitoring (BioButton, CardioWatch 287-2, and C-Med Alpha [10–12]).

Blood pressure can be measured using photoplethysmography (PPG) which is a non-invasive optical technique that detects changes in microvascular blood volume through analysis of light absorption variation. The direct current (DC) component represents baseline tissue absorbance, whereas the alternating current (AC) component reflects cardiac-synchronous arterial blood volume changes. This enables reliable extraction of SpO2, PR, and RR through validated algorithms, with the perfusion index (AC/DC ratio) serving as a clinical indicator of peripheral perfusion status. Waveform morphology analysis permits non-invasive assessment of arterial stiffness through pulse wave features [13]. Wearable devices such as Biobeat [14] and CardioWatch 287−2 [15] have demonstrated blood pressure estimation capabilities through PPG analysis in controlled, calibrated settings, though pulse transit time-based BP estimation encounters substantial technical limitations, including pre-ejection period contamination and insufficient signal information content, with long-term ambulatory accuracy remaining inadequately characterized. The Checkpoint Cardio's CPC12S attempted blood pressure measurement through PPG but demonstrated unacceptably low accuracy in validation testing, exemplifying the current limitations of cuffless BP estimation from optical signals [5].

The Biobeat system (Biobeat Technologies Ltd., Petah Tikva, Israel) features a disposable chest patch or wristwatch that wirelessly calculates an array of vital signs including PR, RR, SpO2, systolic BP (SBP), diastolic BP (DBP), mean BP (MBP), pulse pressure (PP), stroke volume, cardiac output, cardiac index, systemic vascular resistance, and skin temperature. Note that Biobeat does not record HR, which is calculated from an electrocardiogram, but rather PR, which is derived from pulse wave signals obtained via PPG. The Biobeat sensor has been successfully applied in diverse clinical contexts, including the detection of deterioration during the COVID-19 pandemic in both hospitals and home settings [16,17], identification of hypotension during hemodialysis [18], monitoring of maternal hemodynamics during labor [19], and assessment of diuretic responses in congestive heart failure [20]. Recently, Belliveau and colleagues explored its use for continuous monitoring after ambulatory surgery [21].

This multicenter prospective observational study assessed the performance of the Biobeat wearable sensor in postoperative patients monitored for several days without human intervention, focusing on technical performance, reliability, and the device's capability to detect vital sign abnormalities.

## Methods

### Study design, ethics approval and setting

This prospective, multicenter, observational cohort study was conducted at three private, nonprofit hospitals. The study protocol (S1 Appendix) was approved by the Ethical Committee Île-de-France II (no. 2020-A01852-37, September 28, 2020) and registered with ClinicalTrials.gov (NCT04585178, October 14, 2020). Written informed consent was obtained from all participants prior to enrollment, after the study protocol (including duration, methods, and outcome assessment) had been explained and the device demonstrated. The study adhered to the Declaration of Helsinki [22] and followed the Strengthening the Reporting of Observational Studies in Epidemiology (STROBE) guidelines (S2 Appendix) [23].

### Patient population

Adults (≥ 18 years of age), classified as American Society of Anesthesiologists (ASA) physical status 1–3, undergoing major surgery (digestive, gynecological, orthopedic, or urological procedures with expected durations of > 2 hours and a postoperative ward stay of ≥ 2 nights) were eligible regardless of comorbidity. Exclusion criteria included significant

thoracic deformities, skin conditions, allergies to device materials, cardiac pacemakers, tremors, convulsions, pregnancy, breastfeeding, any planned thoracic computed tomography scan or magnetic resonance imaging within 3 days post-surgery. Anesthesia and postoperative analgesia were determined by the attending anesthesiologist, and postoperative care was overseen by the surgeon according to standard protocols.

### Biobeat patch

The Biobeat chest patch, embedded with a sensor, was positioned 1 cm to the left of the sternum, below the clavicle (**S3 Fig**).

The sensor (model BB-613WP) uses a unique reflective PPG technology in which the light source and sensor array are positioned on the same side. Light is transmitted into the subject's skin, and a portion is reflected from the tissue toward a photodiode detector. The PPG signal was collected with high temporal and quantitative resolution. Biobeat devices emit light and analyze reflected signals to detect blood volume changes in the skin. This technology tracks blood pressure changes based on the Pulse Wave Transit Time (PWTT) principle, which measures the time required for a pressure wave from a heartbeat to travel between two arterial points. The device was calibrated using an oscillometric blood pressure monitor to ensure accuracy.

The photoplethysmography-based device has US Food and Drug Administration clearance (FDA) clearance (2019; 510 (K) for measurement of blood pressure, oxygenation, and PR in hospitals, clinics, long-term care facilities, and home settings) and a European Conformity (CE) mark approval (certificate no. 688840; issued March 19, 2019; updated August 29, 2023) for continuous monitoring but not for alerting or alarming.

### Study procedure

Patients received vital sign monitoring through both remote continuous monitoring and routine nurse assessments.

The research staff applied the Biobeat patch in the post-anesthesia care unit and configured the secure cloud-based platform Instamed using patients' unique identifiers and demographic data (age, weight, and height) [24]. Instamed was used only for data storage, and Biobeat Technologies had no access to patient data. BP was recorded using a standard cuff-based device for calibration of Biobeat. Wireless connectivity between the patch and the cloud repository via Wi-Fi router was confirmed before discharge from the post-anesthesia care unit. Automatic connection reestablishment was anticipated when patients transferred to the general surgical ward, where an additional Wi-Fi router was installed in each patient room.

Nurses evaluated the patients 2–3 times daily (morning and evening or morning, midday, and evening) as per standard protocol. They assessed BP and PR using a non-invasive oscillometric monitor. They estimated respiratory rate from 15- or 30-second observation periods and recorded SpO2 and auricular temperature, which was systematically adjusted by +0.5°C to approximate core body temperature.

Monitoring analysis commenced upon ward admission and was planned for a minimum of 48 hours unless terminated early due to patient withdrawal, or necessity for thoracic computed tomography scan, or magnetic resonance imaging. Research assistants conducted daily adverse event checks and removed the patch at the conclusion of the study.

### Data collection

Each patient was assigned a unique identifier at sensor installation. Demographic data and nurse measurements with corresponding timestamps were extracted from hospital medical records. Although Biobeat real-time data display was available, it was not utilized during this study. Biobeat data were retrieved post-study from the Instamed company [24]. We aimed to obtain synchronous measurements of vital signs (PR, MBP, RR, SpO2, and temperature) at 5-minute intervals. Artifact frequency (secondary outcome indicating technical performance) was particularly examined, with objective definitions based on previously published criteria. Artifacts were defined as values exceeding 50% deviation from the prior reading (unless followed by a ±25% recovery value) or falling outside physiological ranges (e.g., PR < 5 or >250 bpm;

systolic BP<20 or >300 mmHg; SpO2 change ≥8% between consecutive readings) [25–28]. After numerical analysis, four independent experts reviewed the tracings to identify additional artifacts not captured by the automated criteria.

Nurses rated the ease of sensor installation on a 4-point Likert scale (0 = no problem, 3 = serious problems). Post-removal skin condition was assessed on a 4-point scale (0 = normal, 3 = severe inflammation), and patients rated their overall satisfaction with the device (0 = very satisfied, 3 = very dissatisfied). Major postoperative complications were recorded at hospital discharge using the Clavien-Dindo classification (grade ≥IIIA defined as major complications [29].

### Outcomes

The primary outcome was the proportion of patients with major hemodynamic abnormalities defined as mean blood pressure (MBP) <60 mmHg, detected by the Biobeat patch. The secondary outcome was the proportion of patients with vital sign abnormalities at various predefined thresholds as detected by either nurse monitoring or the Biobeat device. Other secondary outcomes include data loss and artifact frequency (technical performance), incidence of postoperative complications, assessment of sensor installation ease, and evaluation of patient acceptability and post-removal skin reactions.

### Sample size

This study aimed to recruit 114 participants to ensure approximately 100 would be evaluable, accounting for an anticipated attrition rate of around 12%. This sample size was based on a study published by Liem and colleagues, which reported that postoperative hypotension (for example MBP<60 mmHg) occurred in 8% of surgical patients [30].

### Statistical analysis

For continuous paired measurements, nurse-recorded values and corresponding Biobeat values (recorded within 5 minutes of nurse assessment) were compared using the Bland-Altman method for repeated measures, which evaluates agreement between two measurement methods by plotting differences against means and calculating within- and between-subject variance–adjusted limits of agreement [31]. Comparisons were not performed if the interval between these measurements was greater than 5 minutes. These paired measurements were also evaluated for clinical accuracy using Clarke Error Grid analysis, originally developed for self-monitoring of blood glucose [32]. A standard scatter plot of sensor and nurse recordings was generated and divided into five zones (A through E). Zones A and B indicate clinically acceptable agreement, whereas zones C through E indicate increased discrepancies. Clarke Error Grid zones were defined according to the authors' clinical judgment and criteria from prior validation studies [8,33–35]. For binary paired data, agreement between nurse and Biobeat classifications was evaluated using Cohen's kappa coefficient, which quantifies agreement beyond that expected by chance, though it is sensitive to the distribution of positive and negative categories [36]. To determine whether discordant pairs favored one method (whether nurse or patch produced significantly more positive or negative results), McNemar's test with continuity correction was applied [37]. When the total number of discordant pairs was < 10, the exact McNemar test was used instead.

Only MBP was analyzed and reported for BP, omitting systolic and diastolic values. Results are expressed as numbers (percentages) and medians [25th-75th percentiles]. Statistical analyses were performed using R (version 4.3.0 + ; R Foundation for Statistical Computing, Vienna, Austria). These data are available at Dryad (https://doi.org/10.5061/dryad.69p8cz9dh).

## Results

### Participants' characteristics

From December 15, 2020, to December 14, 2022, 109 postoperative patients were enrolled across the three centers. Biobeat® sensor data were unavailable for 19 patients due to technical issues (n = 17) or patient refusal post-inclusion (n = 2), leaving 90 patients for analysis (69 from Center 1, 5 from Center 2, and 16 from Center 3) (Fig 1).

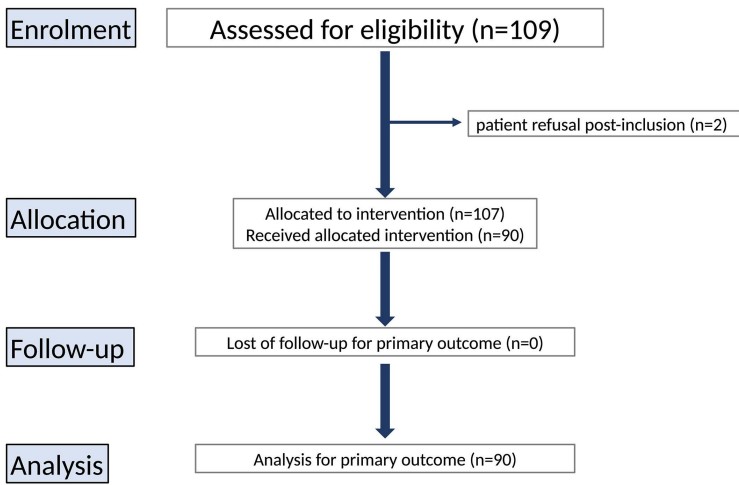

**Fig 1. Flow chart.**

Table 1 summarizes patient demographics and surgical procedures. The monitoring duration, from ward admission to sensor removal, ranged from 12 to 118 hours, with a median of 63 hours [42–72 hours] (Fig 2). All cases of early patch removal (before 48 hours of monitoring) were performed at the patient's request, with the exception of one case in which an unscheduled MRI (magnetic resonance imaging) was required.

## Biobeat monitoring system data collection

Of 64,088 expected simultaneous multi-parameter physiological measurements at 5-minute intervals, 28,090 (43.8%) were missing, while 29,514 contained complete data for all variables. Among the 6,484 incomplete serial measurements, 89.6% lacked $SpO_2$, 11.1% lacked MBP, and 8.9% lacked RR. PR and temperature were missing in only 7 and 3 instances, respectively. Overall, $SpO_2$ was the most frequently absent variable, occurring in 16.1% of all obtained serial measurements, either in isolation or alongside other missing parameters. Fig 3 illustrates an example of complete monitoring and incomplete data with periodic gaps.

## Biobeat sensor artifacts

A total of 230 artifacts were identified using numerical criteria, and 44 additional artifacts were identified through clinical trace review, with artifacts predominantly affecting $SpO_2$, temperature, and RR (**Table 2**).

**Fig 4** displays the distribution of Biobeat-measured parameters across the monitoring period, following artifact exclusion. The median values were: MBP, 89 mmHg [81–97]; PR, 77 bpm [68–89]; RR, 16 breaths/min [14–18]; SpO2, 96% [94–98]; and temperature, 37.8°C [37.5–38.1].

## Comparison of variables captured by biobeat sensor vs. nurses

The analyses included 561 pairs of MBP, 480 pairs of SpO2, 571 pairs of PR, 468 pairs of temperature, and 115 pairs of RR. Bland-Altman analysis revealed small mean biases but wide limits of agreement for all variables. Notable exceptions were respiratory rate (underestimation by Biobeat compared to nurse measurements) and temperature (overestimation by Biobeat compared to nurse measurements) (**Fig 5**). Clarke Error Grid analysis demonstrated excellent concordance for SpO2 (100% in Zone A) and PR (92.8% in Zone A), with lower agreement for MBP (32.4% in Zone B and 3.4% in Zone C), temperature (53.4% in Zone B), and RR (55.6% in Zone B and 4.4% in Zone D) (**Fig 6**).

**Table 1. Patient characteristics and surgical procedures.**

| | Patients included in the analysis n = 90 |
|---|---|
| **Sex, female/male*** | 39 (43.3%)/51 (56.7%) |
| **Age, years**** | 63.5 [51.2–72.7] {32–86} |
| **Body mass index (kg/m²)**** | 25.4 [23.1–28.7] {17.6–56.0} |
| <18.5* | 2 (2.2%) |
| 18.5–24.9* | 39 (43.3%) |
| 25.0–29.9* | 32 (35.6%) |
| >30* | 17 (18.9%) |
| **American Society of Anesthesiologists physical status*** | |
| 1 | 5 (5.6%) |
| 2 | 56 (62.2%) |
| 3 | 29 (32.2%) |
| **Major comorbidities*** | |
| Cardiovascular disease | 46 (51.1%) |
| Hypertension | 38 (42.2%) |
| Ischemic heart disease | 4 (4.4%) |
| Valvular heart disease | 4 (4.4%) |
| Heart failure | 1 (1.1%) |
| Arrhythmia | 11 (12.2%) |
| Peripheral arterial disease | 4 (4.4%) |
| Other cardiovascular disease | 9 (10.0%) |
| Chronic obstructive pulmonary disease | 2 (2.2%) |
| Diabetes | 18 (20.0%) |
| Cancer (in remission or progressive) | 51 (56.7%) |
| **Surgical procedures*** | |
| Digestive | 24 (26.7%) |
| Gynecologic | 16 (17.8%) |
| Urologic | 50 (55.6%) |
| **Surgery length, minutes**** | 191 [152–250] {73–501} |

*: Categorial variables are presented as number (percentage).

**: Continuous variables are presented as median [25th-75th percentiles] {minimal value – maximal value}.

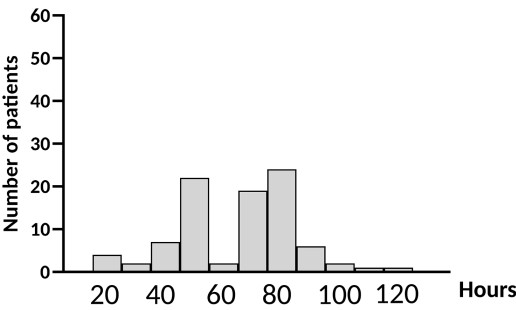

**Fig 2. Distribution of monitoring durations.**

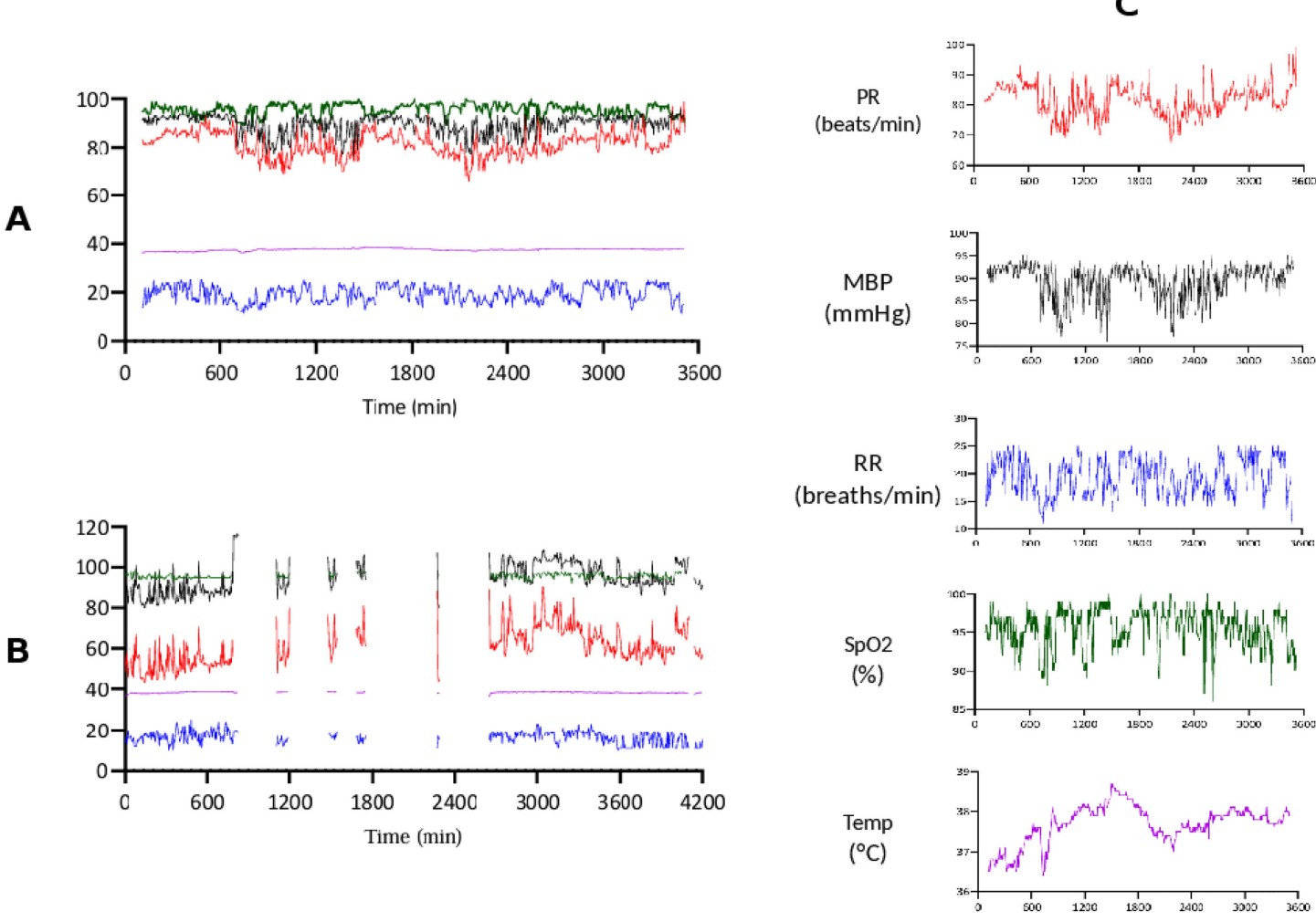

**Fig 3. Example of Biobeat® monitoring.** A: Examples of monitoring without missing data. B: Example of monitoring with missing data. C: Example of individual variable data: PR (pulse rate, beats/min); MBP (mean blood pressure, mmHg); RR (respiratory rate, breaths/min); SpO2 (peripheral oxygen saturation, %); Temp (temperature, °C).

## Variable abnormalities detected by biobeat sensor and nurses

Table 3 presents the hemodynamic, respiratory, and thermal abnormalities detected by Biobeat versus routine monitoring after artifact removal. Agreement between nurse and Biobeat detection of abnormalities was generally modest, with the exception of MBP<60 mmHg, where Cohen's kappa suggested fair agreement (0.30 [−0.04;0.64]). Severe hypotension (MBP<60 mmHg, the primary objective) was not detected more frequently with Biobeat than with nurse surveillance ($P=0.07$, McNemar test), though this difference achieved statistical significance when considering the threshold of MBP<65 mmHg ($P=0.01$). The Biobeat sensor demonstrated significantly higher detection rates for elevated pulse rate (PR≥ 100 beats/min; $P<0.001$), tachypnea (RR>20 breaths/min; $P<0.001$), hypoxemia (SpO2<95%; $P<0.001$), and abnormal temperature (<36.8° or >38 °C; $P<0.001$).

**Table 2. Artifacts detected in Biobeat® data.**

| | Artifacts captured using numerical criteria n (%) | Artifacts based on clinical trace analysis n (%) |
|---|---|---|
| **Pulse rate** | 15 (0.04) | 7 (0.02) |
| **Mean blood pressure** | 13 (0.04) | 18 (0.05) |
| **Respiratory rate** | 48 (0.14) | 13 (0.04) |
| **SpO2** | 93 (0.31) | 4 (0.01) |
| **Temperature** | 61 (0.17) | 36 (0.10) |

Results are reported as number (percentage of total number of measurements).

SpO2: peripheral arterial oxygen saturation.

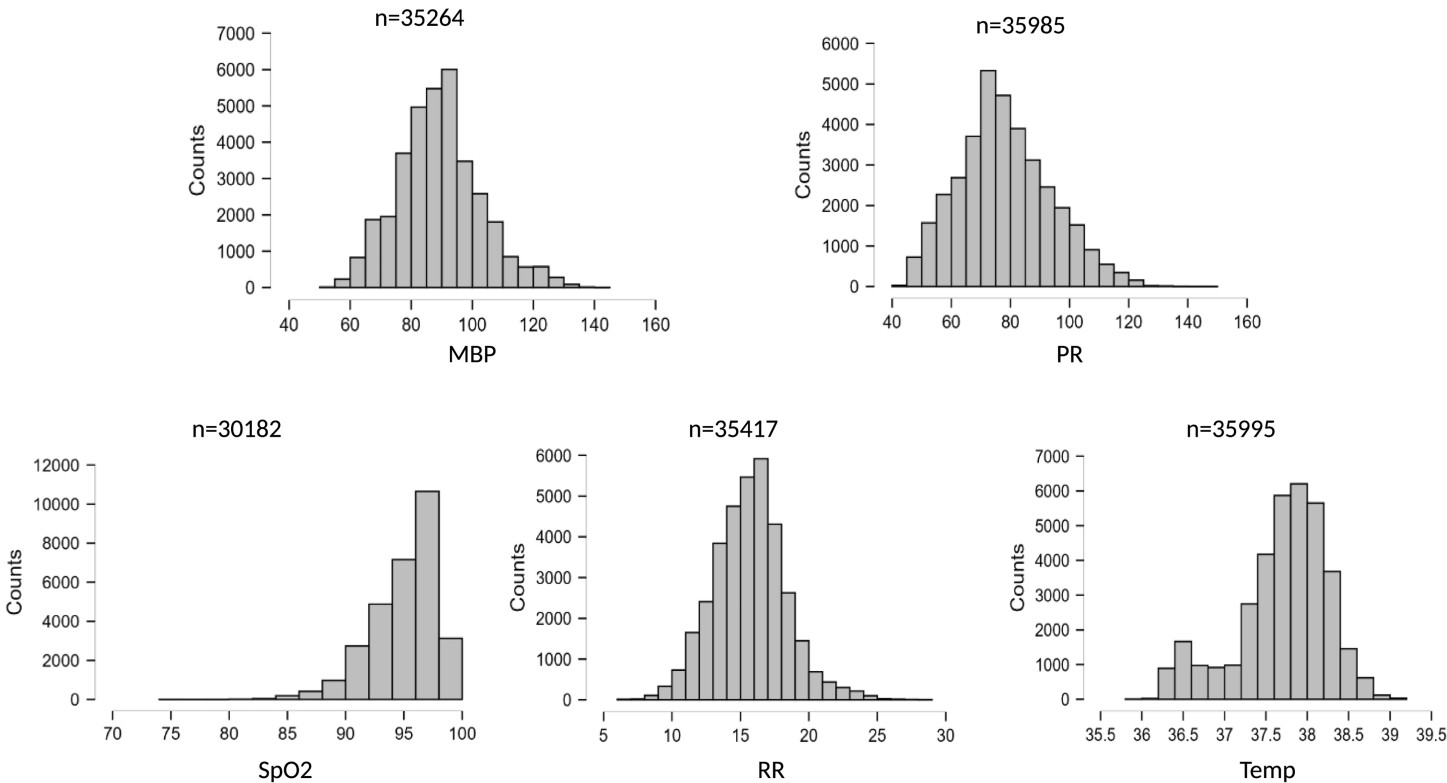

**Fig 4. Distribution of the Biobeat measured parameters across the monitoring period.** n, number of recorded data points; MBP, mean blood pressure (mmHg); PR, pulse rate (beats per minute); SpO2, peripheral oxygen saturation (%); RR, respiratory rate (rate per min); Temp, temperature (°C).

## Other secondary outcomes

Sensor installation was straightforward in 76 cases. Minor difficulties were encountered in 10 patients, and moderate issues occurred in three patients. The sensor was not recognized by the platform in one case, which prevented its configuration in the PACU; the configuration was subsequently completed when the patient transferred to the general surgical ward. Skin assessment at patch removal (n = 87) revealed that 84.4% of the subjects had normal skin and 12.2% had

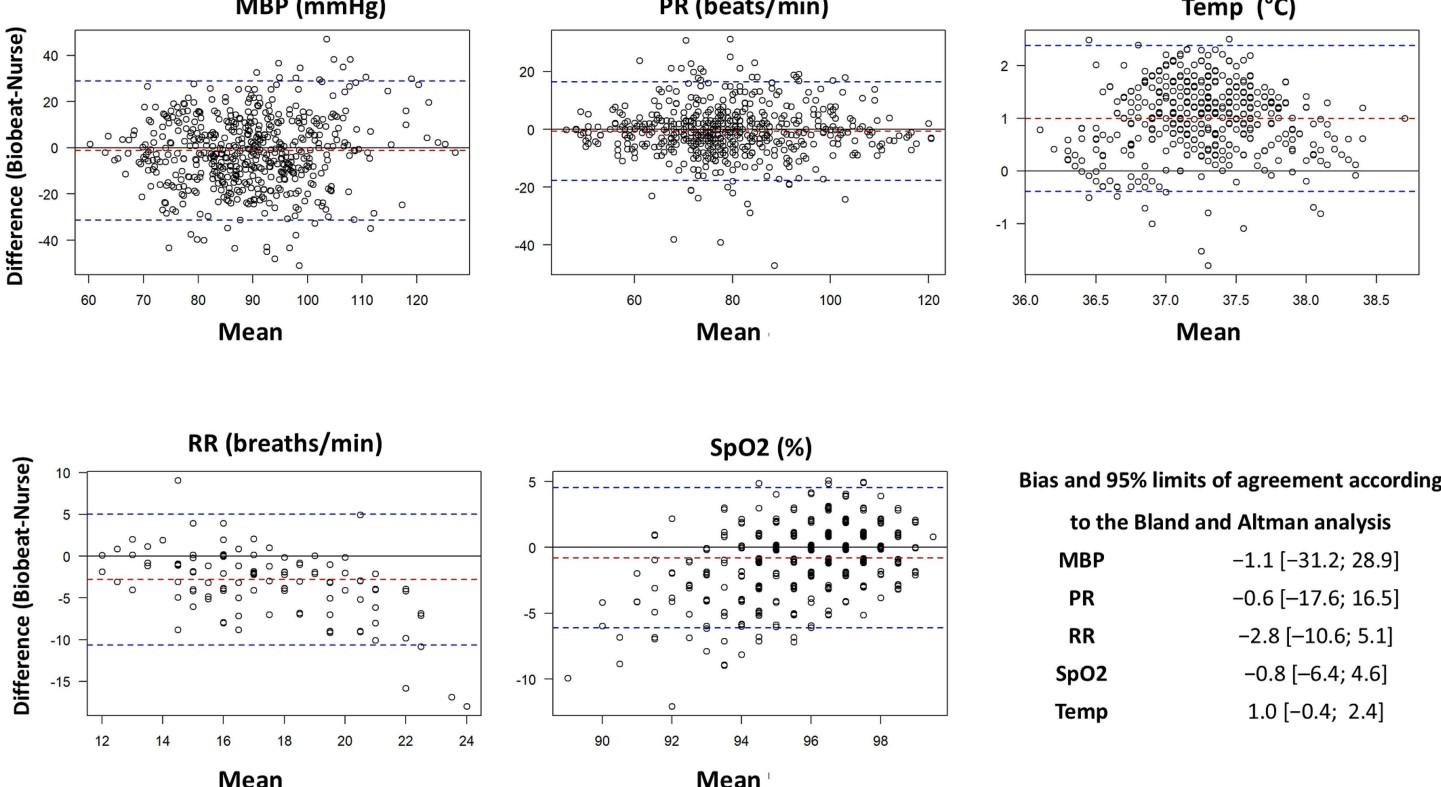

**Fig 5. Bland-Altman comparison of variables captured by Biobeat sensor vs. nurse's measurements.** MBP, mean blood pressure (mmHg); PR, pulse rate (beats per minute); SpO2, peripheral oxygen saturation (%); RR, respiratory rate (per minute); Temp, temperature (°C).

minor reactions. Patient satisfaction (n=88) was high, with 89.8% being very satisfied and 4.5% satisfied; two patients were dissatisfied, and 3 very dissatisfied. No patient experienced major postoperative complications (Clavien-Dindo grade ≥III), although 23 (25.6%) had minor complications.

## Discussion

This study evaluated the Biobeat wearable sensor in postoperative patients, found mixed findings: substantial data loss, infrequent artifacts, and variable agreement with nurse measurements across vital signs parameters. However, the sensor demonstrated a notable ability to identify vital sign abnormalities more frequently than conventional nursing assessments.

### Data loss

A key finding was the substantial loss of monitored data, encompassing both the complete absence of all variables and partial gaps in specific variables. This issue has been well documented. Hara et al. highlighted similar challenges in wireless monitoring [38], while Merschel et al., using a PPG-equipped smartwatch (Cardio Watch 287), reported analyzable data ranging from 34% to 100%, which decreased during activity [39]. Using Biobeat chest patches, Belliveau et al. used Bluetooth and patients' phones and noted that all participants experienced at least one 2-hour data gap [21]. Helmer et al. used Biobeat wrist-worn devices and equipment of patient room and corridors with dedicated routers and reported valid data for 61.5% of the monitored time (range: 20.1–78.3%) [40]. In the absence of data transfer monitoring, we can only speculate regarding the multifactorial causes of these data losses throughout the transmission chain specifically: sensor

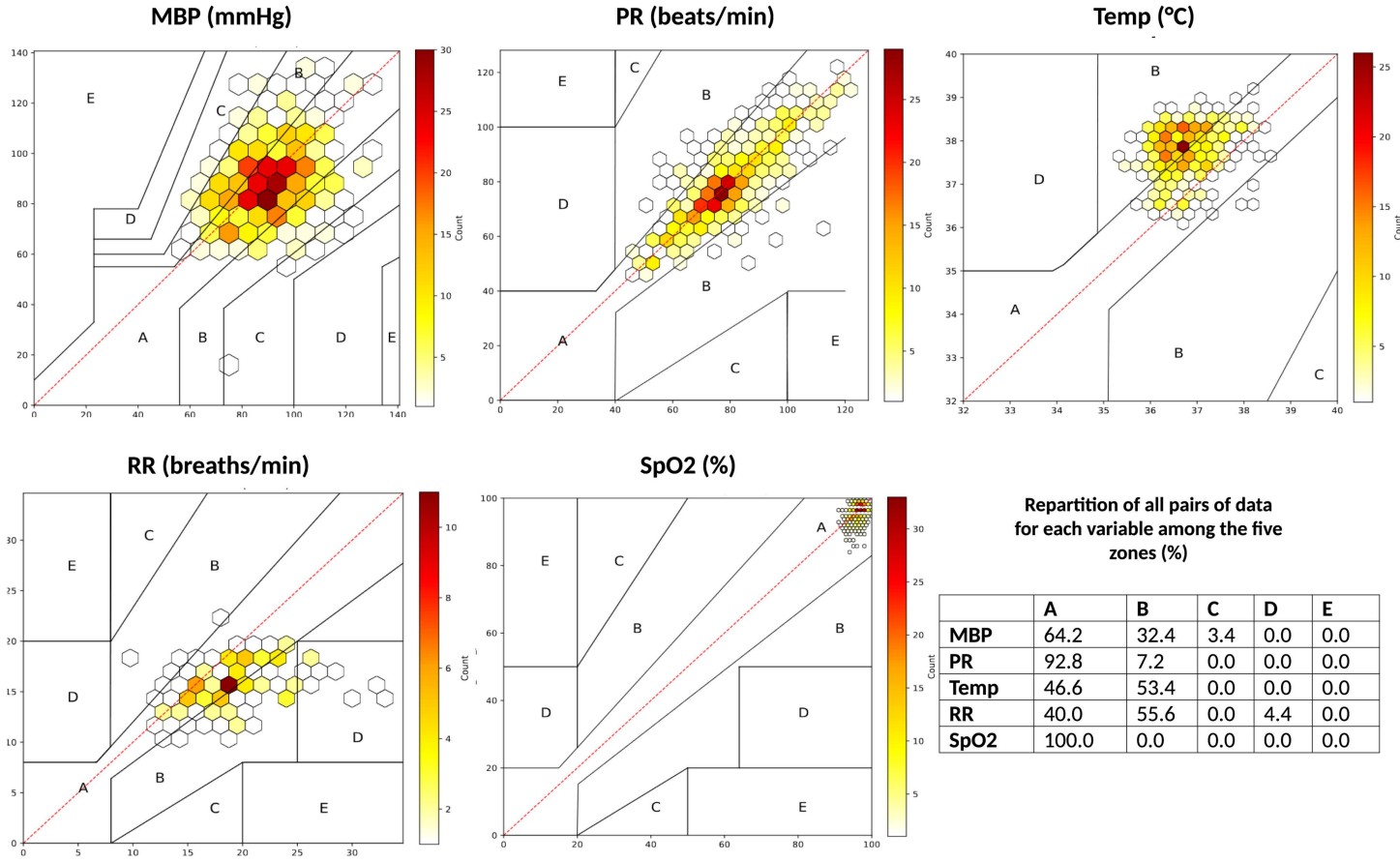

**Fig 6. Clarke Error Grid analysis of variables captured by Biobeat sensor vs. nurse's measurements.** MBP, mean blood pressure (mmHg); PR, pulse rate (beats per minute); SpO2, peripheral oxygen saturation (%); RR, respiratory rate (per minute); Temp, temperature (°C).

detachment (possibly from sweating), the device's internal algorithm rejecting low-quality signals, router or battery limitations, disrupted cloud transfers, and patient displacement beyond the area covered by the router.

Finally, one particular point concerns SpO2, which emerged as the parameter most frequently missing when data transmission has been satisfactory. SpO2 is intrinsically difficult to measure accurately because the arterial pulsatile component of PPG represents only a small fraction of the overall optical signal and is easily contaminated by noise. Motion artifacts, low peripheral perfusion, vasoconstriction, ambient light, and sensor misplacement can all distort the PPG waveform and lead to erroneous SpO2 estimates which are rejected by the device [41].

## Artifact

Multiple elements can affect PPG, including individual variations (e.g., skin tone, obesity, age, and sex), physiological processes (e.g., respiratory rate, venous pulsations, local body temperature, and body site), and external factors (e.g., motion artifacts, ambient light, and applied pressure) [42]. These factors can decrease or modify the PPG intensity or waveform. Despite these potential issues, very few artifacts were detected in our study based on both arbitrary definitions and expert opinions, indicating a very low risk of over-alerting the system. This is critical, as "alarm fatigue" can desensitize staff to alerts [43] and increase patient risk [44]. Edgar et al. used a PPG wristband for cardiac arrest detection and reported minimal false alarms during high-risk procedures [45]. Comparing our findings with those of

**Table 3. Numbers (percentages) of patients with hemodynamic, respiratory, and temperature abnormalities detected by Biobeat or nurses' measurements.**

| | Agreement | Disagreement | | Cohen's Kappa coefficient | McNemar test P value |
|---|---|---|---|---|---|
| | N+/ B+ and N-/ B- | N+/ B- | N-/ B+ | | |
| **Mean blood pressure (mmHg%)*** | | | | | |
| <60 | 82 (91.1%) | 1 (1.1%) | 7 (7.8%) | 0.30 [−0.04;0.64] | 0.070 |
| <65 | 75 (83.3%) | 2 (2.2%) | 13 (14.4%) | 0.22 [−0.03;0.47] | **0.010** |
| <70 | 68 (75.6%) | 11 (12.2%) | 11 (12.2%) | 0.24 [0.00;0.47] | 1.000 |
| >100 | 51 (56.7%) | 21 (23.3%) | 18 (20.0%) | 0.12 [−0.09;0.32] | 0.749 |
| >110 | 58 (64.4%) | 18 (20.0%) | 14 (15.6%) | 0.17 [−0.04;0.38] | 0.596 |
| >120 | 76 (84.4%) | 6 (6.7%) | 8 (8.9%) | 0.04 [−0.20;0.28] | 0.789 |
| **Pulse rate (beat/min%)*** | | | | | |
| ≤40 | 88 (97.8%) | 0 (0.0%) | 2 (2.2%) | – | 0.500 |
| ≥100 | 46 (51.1%) | 5 (5.6%) | 39 (43.3%) | 0.13 [−0.01;0.28] | **< 0.001** |
| **Respiratory rate (breath/min%)*** | | | | | |
| <8 | 87 (96.7%) | 0 (0.0%) | 3 (3.3%) | – | 0.250 |
| >20 | 36 (40.0%) | 4 (4.4%) | 50 (55.6%) | 0.00 [−0.11; 0.11] | **< 0.001** |
| **Peripheral oxygen saturation (%)**** | | | | | |
| <95 | 46 (51.7%) | 4 (4.5%) | 39 (43.8%) | 0.11 [−0.03;0.25] | **< 0.001** |
| <92 | 38 (42.7%) | 0 (0.0%) | 51 (57.3%) | 0.06 [0.00;0.11] | **< 0.001** |
| <90 | 55 (61.8%) | 0 (0.0%) | 34 (38.2%) | 0.03 [−0.03;0.10] | **< 0.001** |
| <85 | 84 (94.4%) | 0 (0.0%) | 5 (5.6%) | – | 0.063 |
| **Temperature (°C)*** | | | | | |
| <36.8° | 59 (65.6%) | 30 (33.3%) | 1 (1.1%) | 0.02 [−0.07; 0.11] | **< 0.001** |
| >38° | 34 (37.8%) | 1 (1.1%) | 55 (61.1%) | 0.06 [−0.02;0.13] | **< 0.001** |
| >39° | 86 (95.6%) | 0 (0.0%) | 4 (4.4%) | 0.00 [0.00; 0.00] | 0.125 |

Kappa coefficient is reported as median [25th-75th percentiles].

N+: number of patients (%) for whom an abnormality was found during nursing monitoring.

N-: number of patients (%) for whom no abnormality was found during nursing monitoring.

B+: number of patients (%) for whom an abnormality was found by Biobeat.

B-: number of patients (%) for whom no abnormality was found by Biobeat.

*: 90 patients.

**: 89 patients.

other studies is challenging because of inconsistent artifact definitions and an unclear boundary between abnormal vital signs and artifacts. Such comparisons are also complicated by differences in methodology (for example, per-measurement analysis versus moving medians [46] or Butterworth low-pass filters [35], which may obscure sudden changes in vital signs.

## Reliability of Biobeat measurements

For MBP, our results differ little from those obtained by the Checkpoint Cardio's CPC12S [5] with large limits of agreement (Bland-Altman analysis) and some pairs falling into Zone C (Clarke Error Grid), signaling potential over- or undertreatment risks. However, previous studies that specifically analyzed the measurement of blood pressure using the Biobeat sensor have shown better results compared with the reference method (sphygmomanometry) [14] and invasive

methods [47]. Furthermore, van Vliet et al. provide results supporting the feasibility and accuracy of PPG-based BP measurements using the CardioWatch 287−2 in patients undergoing heart catheterization [15].

The Biobeat PR measurements showed favorable results. However, Xu et al. reported that only 74.5% of the HR measurements from the Empatica E4 wrist sensor were within ± 10 bpm of the reference standard [35].

All SpO2 measurement pairs were in Zone A, but a few patients experienced low SpO2, similar to the NIGHTINGALE clinical validation study [5], making the study of the precision of this variable questionable.

RR pairs occasionally reached Zone D, in contrast to the findings of Eisenkraft et al. [48]. Methodological differences complicate direct comparisons with other Biobeat studies [14,47,49] or devices such as EarlySense [50], SensiumVitals [51], SmartCardia [33], and CardioWatch 287−2 [11]. Nonetheless, our findings align with those of HealthPatch [8,28], SensiumVitals [51], and VitalPatch [52].

The Biobeat temperature measurements overestimated the nurse-recorded values, with pairs nearly evenly split between zones A and B. Xu et al. also noted moderate precision and wide limits of agreement, with only 29.6% of wrist sensor temperature readings being within ±1°C of the reference standard [34].

Overall, the sensor accuracy warrants substantial caution, varying according to vital signs.

### Ability to detect vital sign abnormalities relative to standard care

The results suggest the value of continuous wireless monitoring, pending confirmation by prospective trials, such as Jensen et al., who linked it to the earlier detection of serious events [53]. This value could be even more important in high-risk patients (older age, comorbidities, and some types of surgical procedures). This was not the case in our study, as shown by the low number of patients, or even their absence, when considering the occurrence of very severe hypotension (MBP < 60 mmHg), bradycardia (PR ≤ 40 beats/min), bradypnea (RR < 8 breaths/min), or hyperthermia (temperature > 39°C).

### Patient satisfaction

Patients reported high satisfaction with Biobeat, echoing SensiumVitals findings [8], although some withdrew, likely feeling well, and perceived little benefit from wireless monitoring.

### Strengths and limitations

A major strength of this study was the evaluation of the feasibility and reliability of a transcutaneous device for measuring vital signs, including blood pressure, in real-word ward conditions. However, this study has several limitations.

First, we chose the Biobeat precordial sensor over the wrist monitor because of its simplicity, cost-effectiveness, and infection control benefits. This decision restricted patient mobility, an issue we had not foreseen, as we omitted real-time data flow monitoring and caregiver alerts for interruptions. A potential solution, as demonstrated by Breteler et al. in post-esophagectomy patients [52] and Belliveau et al. after ambulatory surgery [21], is to connect directly to the patient's mobile phone. However, Belliveau et al. reported data loss using Biobeat. Alternative methods, such as low-power wireless protocols with hotspot connectivity and centralized server architecture [53], present trade-offs worth investigating [54].

Second, the recruitment targeted 114 participants but fell to 90 due to changes in recruitment and practices at the time of the COVID-19 epidemic, which nevertheless revealed reliability issues.

Third, we included patients with an expected postoperative hospital stay of ≥ 2 nights. This criterion was based on evidence that the majority of postoperative complications (approximately 60%) occur within 1–3 days after surgery, as demonstrated in a cohort study of patients undergoing intra-abdominal operations [55]. However, future prospective studies should evaluate extended monitoring in patients requiring prolonged postoperative supervision (e.g., those undergoing complex surgeries or with severe comorbidities), with monitoring extending at least through the first week post-discharge.

Fourth, we used numerical and clinical methods to detect artifacts. The numerical evaluation takes each value into account and can isolate abnormal values (i.e., a single value between two periods where this variable is not measured), while the clinical evaluation considers all the data and can therefore detect an anomaly based on the context and trend of the variable). This explains, at least in part, the results observed. However, these approaches did not account for the Biobeat internal algorithms. Furthermore, machine learning approaches, demonstrated in ICU vital sign monitoring applications [56], could improve artifact detection and enhance clinical relevance.

Fifth, Clarke Error Grid analysis, which aims to provide a patient safety-focused framework for evaluating measurement accuracy, has notable limitations. The boundaries of its zones (A to E) are based on clinical assumptions that remain debatable. For example, Saugel et al. defined BP risk zones using a survey of 25 anesthesiology and intensive care experts [57]. Furthermore, the number of zones should be reconsidered, potentially including a safe zone, a possible risk zone, and a definite abnormality zone. This is particularly relevant for measurements such as peripheral oxygen saturation, which has an approximate precision of ±2%.

Sixth, comparing Biobeat data with data recorded by nurses requires time alignment, which can be imperfect within a few minutes of recording. Nurses often recorded measurements with delays in their software, whereas Biobeat was recorded every five minutes. Additionally, the RR measured by the nurse, usually extrapolated from 15- or 30-second counts, is error-prone [51]. Furthermore, accurate blood pressure measurement requires proper calibration using a standard cuff-based device, as the device estimates pressure indirectly from optical signals rather than measuring it directly. Moreover, the conditions under which the calibration was performed may have influenced the quality of the subsequent measurements.

Seventh, we did not calculate the Early Warning Score (EWS) [58], as Biobeat lacks data on oxygen administration and neurological status, which are critical for detecting postoperative brain dysfunction, particularly in frail patients. Alternatively, we could have used a Biobeat-adapted multiparameter real-time warning score (MPRT-WS) incorporing PR, SpO2, RR, SBP, DBP, temperature, stroke volume, cardiac output, and systemic vascular resistance [59].

## Generalizability

This study can be replicated without difficulty

## Conclusion

Ensuring reliable data transmission is critical for minimizing the impact of missing data on the analysis. Further studies are needed to validate the accuracy of the monitored variables, especially blood pressure and respiratory rate, which are essential for the early detection of postoperative decline and timely intervention [60]. The rise of continuous remote monitoring systems underscores the need for independent validation conducted under standard clinical conditions and free from manufacturer bias, as startups, medical device companies, and software firms compete in this expanding market.

## Supporting information

**S1 Appendix. Protocol summary.**
(PDF)

**S2 Appendix. Strobe checklist.**
(DOCX)

**S3 Fig. Biobeat sensor and its positioning [from Paternot et al. F1000Res.** 2021;10:622. https://doi.org/10.12688/f1000research.54781.2 with permission].
(EPS)

## Acknowledgments

The authors would like to thank Jérémy Neuberg for allowing us to use the Instamed platform and also like to thank Polly Gobin for her linguistic assistance.

## Author contributions

**Conceptualization:** Aurélie Blondeau-Martin, Karine Merlin, Bernard Trillat, Morgan Le Guen, Marc Fischler.

**Data curation:** Cécile Landais, Audrey Solis, Delphine Petit, Aurélie Blondeau-Martin, Karine Merlin.

**Formal analysis:** Inès Kouidri, Cécile Landais, Marc Fischler.

**Funding acquisition:** Marc Fischler.

**Investigation:** Inès Kouidri, Audrey Solis, Delphine Petit, Aurélie Blondeau-Martin, Karine Merlin.

**Methodology:** Morgan Le Guen, Marc Fischler.

**Project administration:** Morgan Le Guen, Marc Fischler.

**Supervision:** Marc Fischler.

**Visualization:** Inès Kouidri, Audrey Solis, Delphine Petit, Aurélie Blondeau-Martin, Karine Merlin, Bernard Trillat.

**Writing – original draft:** Inès Kouidri, Cécile Landais, Morgan Le Guen, Marc Fischler.

**Writing – review & editing:** Inès Kouidri, Cécile Landais, Audrey Solis, Delphine Petit, Aurélie Blondeau-Martin, Karine Merlin, Bernard Trillat, Morgan Le Guen, Marc Fischler.

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
