## [Decision Letter · Decision Letter 0]

29 Oct 2025

Dear Dr. Fischler,

Thank you for submitting your manuscript to PLOS ONE. After careful consideration, we feel that it has merit but does not fully meet PLOS ONE’s publication criteria as it currently stands. Therefore, we invite you to submit a revised version of the manuscript that addresses the points raised during the review process.

We look forward to receiving your revised manuscript.

Kind regards,

Agnese Sbrollini

Academic Editor

PLOS ONE

Journal Requirements:

2. We note that there is identifying data in the Supporting Information file <Protocol-Summary-French and English translation.pdf, Ethical Commitee Approval-French and English translation.pdf>. Due to the inclusion of these potentially identifying data, we have removed this file from your file inventory. Prior to sharing human research participant data, authors should consult with an ethics committee to ensure data are shared in accordance with participant consent and all applicable local laws.

-Location data

3. We are unable to open your Supporting Information file [S4 Fig 1 – Riviere – BioBeat device.eps]. Please kindly revise as necessary and re-upload.

Reviewers' comments:

Reviewer's Responses to Questions

**Comments to the Author**

1. Is the manuscript technically sound, and do the data support the conclusions?

Reviewer #1: Partly

Reviewer #2: Partly

2. Has the statistical analysis been performed appropriately and rigorously?

Reviewer #1: Yes

Reviewer #2: N/A

3. Have the authors made all data underlying the findings in their manuscript fully available?

Reviewer #1: Yes

Reviewer #2: No

4. Is the manuscript presented in an intelligible fashion and written in standard English?

Reviewer #1: Yes

Reviewer #2: Yes

Reviewer #1: Minor comments

Line 88: More inclusion criteria is to be provided.

Line 151: This statement should not be placed in the statistical analyses section but a subtitle under ‘sample size’. The statement could be improved. e.g. We aimed to recruit 114 participants to ensure that approximately 100 would be evaluable, accounting for an anticipated attrition rate of about 12–13%. This sample size was considered sufficient a priori for a pilot study.

Line 164: The sentence ‘Paired nominal data were compared using the McNemar’s test’ requires revision. e.g. Paired nominal data were compared using McNemar’s test.

Table 1 Age, years, BMI: Indicate what the figures refer to.

Line 185: To use symbols to indicate number (percentage) and median [25th and 75th percentiles] {minimal and maximal values} in the table and denoted in the table footnote/

Table 2: n(%) is to be placed on top of the table after the column variable name.

Table 3, Line 246: { } is to be replaced with ( ). The statistical test is to be denoted in the table footnotes.

S4 Figure 1: The figure is inaccessible.

The manuscript contains a few typographical/grammatical errors and would benefit from thorough proofreading.

Reviewer #2: The paper reports the validation of the Biobeat device in postoperative patients. The paper is overall well written. However, there are sections, mostly related with the protocol, which are not clear. For example, I was not able to understand if this is actually a prospective study, or a secondary study that came up later or in parallel. The study protocol is a clinical protocol, while this study is about the Biobeat. Please find below a list of specific comments.

-Comments related to how the study protocol is presented:

--I suggest the authors to add a few details of the population within the paper and not only in the supplementary material. For example, what type of surgery was considered.

--The study has, as a primary objective, the estimate of the proportion of patients with severe hypotension, and several secondary objectives. Is this study related to only the secondary objectives? Or, the results of the study protocols are already published and this is a retrospective (and not prospective) analysis? I highly suggest authors to clarify the context.

--The sentence "We aimed to recruit patients to yield evaluable participants, which is a sufficient number a priori for a pilot study" is not scientifically supported. Why pilot study? The statistical analysis was already done. Again, are authors reusing data collected for another study? If so, the narrative must be completely changed!

-I am not clear with the protocol. It is said that the number of nights for the post-operative stay was >=2. However, the monitoring was done after ward admission for 72 hours. Was the monitoring concluded by the research assistants if the number of nights was exactly 2 (so, less than 72 h)?

-Please provide the sampling rate and amplitude resolution of the device.

-Please provide whether the experts validated the artifacts automatically detected or not. From the description "four experts reviewed the tracings to identify other artifacts", it seems they only added additional artifacts. However, from Table 2, the artifacts from clinical inspection were always lower and automatic rules. Please clarify.

-In the sentence "Biobeat does not include an electrocardiogram and therefore measures PR and not HR [11]", is [11] an appropriate citation? It looks as a study in a similar domain but not necessarily supporting that you cannot estimate HR without ECG. Please, clarify.

-How the time of measurements by the nurses was recorded?

-Double check the minimal resolution of the images for publishing with this journal. On my laptop, the image "S4 Fig 1 - Riviere - BioBeat device" looks rather course. Figure 3 looks course too.

-Is there a typo in Fig. 5? The CI for Temp is all negative but the bias is positive.

-I am not understanding the values in curly brackets in Table 3. What is "Number of observations are reported as {}"? Observation=patients?

-Table 3 states reporting the number of patients. However, the percentages are not summing up to 1. I think authors are reporting values across measurements and not patients.

-What are the "1.1% serious issues" in sensor installation?

-What data support "the sensor outperformed conventional monitoring in detecting vital sign abnormalities" in this study?

-I am not sure what authors mean with "real-world" study. The clinical protocol refers to a clinical study (which is of course real). In most studies involving wearables, "real-world" means "into the wild" ie outside the clinical environment, where there is little to no control of what the person is doing during monitoring. Please, clarify.

-I think authors should give some additional details to better understand "The simplest hypothesis is that patients move beyond the router’s 10-meter range.". For example: Does the patch have storing capabilities? Does the patch contain a log of connections/disconnections?

-What is LoA at line 307?

-What was the percentage of PR measurements which were within +/- 10 bpm? Figure 5 reports the limits where approximately 20 bpm. So, is it the Biobeat similar to Empatica E4 and other devices if the same threshold is considered?

-The sentence "Finally, the Biobeat® performance has likely improved since the period of this study" should be removed. It does not add anything to the study and it is not supported either by additional references.

-Typos:

--Do not reinitialize acronyms (e.g. PACU)

--Artifacts or Artefacts? Please uniform the use.

110 { 90}

--hadn’t -> had not

.

Reviewer #1: No

Reviewer #2: No

---

## [Author Response · Author response to Decision Letter 1]

8 Dec 2025

Responses to the Reviewers

We would like to thank the reviewers for their comments. Below, we address each point raised, indicating how changes have been incorporated.

In addition, we clarified the comparison of Biobeat versus routine monitoring of detection of abnormalities by including the concordance test. The tests used are explained in the statistics chapter.

Reviewer 1

Comment 1: Line 88: More inclusion criteria is to be provided.

Response to comment 1: We specified more precisely the inclusion criteria as follows: "Adults (≥ 18 years of age) undergoing major surgery (digestive, gynecological, orthopedic, or urological surgery with expected durations of > 2 hours and a postoperative stay of ≥ 2 nights) were eligible.”

Comment 2: Line 151: This statement should not be placed in the statistical analyses section but a subtitle under ‘sample size’. The statement could be improved. e.g. We aimed to recruit 114 participants to ensure that approximately 100 would be evaluable, accounting for an anticipated attrition rate of about 12–13%. This sample size was considered sufficient a priori for a pilot study.

Response to comment 2: We have created a specific section on the number of patients to be included intitled “Sample size”. In addition, the term “pilot study” has been removed as it creates confusion, as pointed out by Reviewer 2.

Comment 3: Line 164: The sentence ‘Paired nominal data were compared using the McNemar’s test’ requires revision. e.g. Paired nominal data were compared using McNemar’s test.

Response to comment 3: Changes have been made as suggested by the reviewer.

Comment 4: Table 1 Age, years, BMI: Indicate what the figures refer to.

Comment 5: Line 185: To use symbols to indicate number (percentage) and median [25th and 75th percentiles] {minimal and maximal values} in the table and denoted in the table footnote.

Response to comments 4 and 5: We have modified Table 1 using symbols and footnotes to summarize the main characteristics of the patients.

*: Categorial variables are presented as number (percentage)

**: Continuous variables are presented as median [25th-75th percentiles] {minimal value - maximal value}.

Comment 6: Table 2: n(%) is to be placed on top of the table after the column variable name.

Response to comment 6: This has been done as suggested.

Comment 7: Table 3, Line 246: { } is to be replaced with ( ). The statistical test is to be denoted in the table footnotes.

Response to comment 7: Table 3 has been redone in accordance with this comment and that of the Reviewer 2.

Comment 8: S4 Figure 1: The figure is inaccessible.

Response to comment 8: We apologize for the incorrect naming of the attached files (Supporting Information).

The corrected version includes the following changes:

S1 Appendix: Protocol Summary

S2 Appendix: Ethical Committee Approval

S3 Appendix: Strobe checklist

S4 Figure: Biobeat sensor and its positioning [from Paternot et al. F1000Res. 2021;10:622. doi: 10.12688/f1000research.54781.2 with permission]

Comment 9: The manuscript contains a few typographical/grammatical errors and would benefit from thorough proofreading.

Response to comment 9: The manuscript has been reviewed, and we hope that errors have been corrected.

Reviewer 2

General comment: The paper reports the validation of the Biobeat device in postoperative patients. The paper is overall well written. However, there are sections, mostly related with the protocol, which are not clear. For example, I was not able to understand if this is actually a prospective study, or a secondary study that came up later or in parallel. The study protocol is a clinical protocol, while this study is about the Biobeat. Please find below a list of specific comments.

to general comment: This is a prospective clinical study. We acknowledge that this was not sufficiently clear in the manuscript. The term “prospective” has been added to the title, abstract, and end of the Introduction section.

However, we believe that the chapter “Study design, ethics approval and setting” provides a sufficient description of the method, especially since we have provided a summary of the protocol (S1 Appendix) and the advice of the ethics committee (S2 Appendix).

Finally, this protocol is a clinical study, even though it concerns medical equipment, as it is in accordance with French legislation at the time of registration.

Comment 1: - Comments related to how the study protocol is presented:

--I suggest the authors to add a few details of the population within the paper and not only in the supplementary material. For example, what type of surgery was considered.

--The study has, as a primary objective, the estimate of the proportion of patients with severe hypotension, and several secondary objectives. Is this study related to only the secondary objectives? Or, the results of the study protocols are already published and this is a retrospective (and not prospective) analysis? I highly suggest authors to clarify the context.

--The sentence "We aimed to recruit patients to yield evaluable participants, which is a sufficient number a priori for a pilot study" is not scientifically supported. Why pilot study? The statistical analysis was already done. Again, are authors reusing data collected for another study? If so, the narrative must be completely changed!

-I am not clear with the protocol. It is said that the number of nights for the post-operative stay was >=2. However, the monitoring was done after ward admission for 72 hours. Was the monitoring concluded by the research assistants if the number of nights was exactly 2 (so, less than 72 h)?

Response to comment 1:

--As suggested also by Reviewer 1, we specified more precisely the inclusion criteria as follows: "Adults (≥ 18 years of age) undergoing major surgery (digestive, gynecological, orthopedic, or urological surgery with expected durations of > 2 hours and a postoperative stay of ≥ 2 nights) were eligible.”

-- We were awkward in describing the outcomes. The revised version is as follows: “The primary outcome was the proportion of patients with major hemodynamic abnormalities (MBP < 60 mmHg) using the data obtained from the Biobeat patch. The secondary outcome was the proportion of patients with vital sign abnormalities at several thresholds, either through nurse monitoring or using the study device. Other secondary outcomes concern the lack of data collection and artifact frequency (technical performance), the occurrence of postoperative complications, evaluation of ease of sensor installation, and evaluation of patients' acceptability and post-removal skin abnormality.”

-- This comment is similar to comment 2 from Reviewer 1. We created a specific section concerning the number of patients to be included in the study. The term “pilot study” has been removed as it creates confusion as pointed by the Reviewer and added the fact that the number of patients to be included was choose arbitrary: “We aimed to recruit 114 participants to ensure that approximately 100 would be evaluable, accounting for an anticipated attrition rate of about 12–13%. This number was chosen from a study published by Liem et al., who reported that postoperative hypotension (for example MBP less than 60 mmHg) occurred in 8% of the patients [17].” This was specified in the protocol (see S1 Annex) but was omitted from the text. As this was a prospective clinical study, the number of patients to be recruited was planned before the study began.

- The paragraph concerning the study duration is poorly worded. We decided on a minimum duration (see inclusion criteria) but not an upper limit of the intervention. The corrected paragraph is as follows: “Monitoring analysis began upon ward admission and was planned for at least 48 h unless terminated early due to patient withdrawal or thoracic computed tomography scan, or magnetic resonance imaging necessity. The research assistants checked for adverse events daily and removed the patch at the conclusion of the study.”

Comment 2: Please provide the sampling rate and amplitude resolution of the device.

Response to comment 2: We found no data on Biobeat’s sampling frequency or amplitude resolution. Similar wearable PPG devices sample raw signals at 50–100 Hz (based on reviews of PPG technology in wearables) to capture pulse waves, with an amplitude resolution of 12–16 bits for a sufficient dynamic range in light intensity measurements. Consequently, a scientifically rigorous and precise answer regarding the sampling and amplitude resolution specifications of the Biobeat device cannot be provided.

Comment 3: Please provide whether the experts validated the artifacts automatically detected or not. From the description "four experts reviewed the tracings to identify other artifacts", it seems they only added additional artifacts. However, from Table 2, the artifacts from clinical inspection were always lower and automatic rules. Please clarify.

Response to comment 3: The search for artifacts was conducted in two ways: by comparing values with threshold values (mathematical" evaluation) and clinical evaluation, as described in the manuscript. The "mathematical" evaluation considers each value and can isolate abnormal values (i.e., a single value between two periods where this variable is not measured), while the clinical evaluation considers all the data and can therefore detect an anomaly based on the context (evolution of the variable). This explains, at least in part, the observed results. We have changed our text to explain this point as follows: “The numerical evaluation takes each value into account and can isolate abnormal values (i.e., a single value between two periods where this variable is not measured), while the clinical evaluation considers all the data and can therefore detect an anomaly based on the context (evolution of the variable). This explains, at least in part, the results observed.”

Comment 4: In the sentence "Biobeat does not include an electrocardiogram and therefore measures PR and not HR [11]", is [11] an appropriate citation? It looks as a study in a similar domain but not necessarily supporting that you cannot estimate HR without ECG. Please, clarify.

Response to comment 4: We apologize for using reference [11], which is not appropriate for this study. We clarified this point at the end of the Biobeat presentation (Introduction section): “Biobeat does not record HR which is calculated from an electrocardiogram but PR which is derived from pulse wave signals, obtained via PPG.” No reference is needed to support this statement.

Comment 5: How the time of measurements by the nurses was recorded?

Response to comment 5: This is specified in the corrected version: “Demographic data and nurse measurements with their measurement times were extracted from hospital records.”

Comment 6: Double check the minimal resolution of the images for publishing with this journal. On my laptop, the image "S4 Fig 1 - Riviere - BioBeat device" looks rather course. Figure 3 looks course too.

Response to comment 6: The resolution of the images was checked. Figure 3 has been redrawn.

Comment 7: Is there a typo in Fig. 5? The CI for Temp is all negative but the bias is positive.

Response to comment 7: We apologize for this typographical error. Ci for Temp is [-0.4; 2.4]. Figure 5 has been modified accordingly in the revised manuscript.

Comment 8: I am not understanding the values in curly brackets in Table 3. What is "Number of observations are reported as {}"? Observation=patients?

Comment 9: Table 3 states reporting the number of patients. However, the percentages are not summing up to 1. I think authors are reporting values across measurements and not patients.

Responses to comments 8 and 9:

Table 3 shows the number of patients and their percentage relative to the number of patients for whom the data were available. The sum is not 100% because it is a "Russian doll" presentation, which is a nested structure. Furthermore, the same patient may be counted in several categories; a patient may have been hypothermic and febrile during the monitoring period, for example.

Table 3 has been modified significantly:

- We have simplified Table 3 to improve clarity. We removed the data concerning the percentage change in blood pressure relative to the value measured before anesthesia, since these data did not add much value.

- We have introduced agreement between by Biobeat and/or by nurses’ measurements of hemodynamic, respiratory, and temperature abnormalities for each threshold.

This leads to two statistical analyses as explained in the statistical chapter: “For binary paired data, agreement between nurse and Biobeat classifications was evaluated using Cohen’s kappa coefficient, which quantifies agreement beyond chance but is sensitive to the distribution of positive and negative categories. To assess whether discordant pairs favored one method (whether nurse or patch produces significantly more positive, or negative results than the other), McNemar’s test with continuity correction was applied. When the total number of discordant pairs was <10, the exact McNemar test was used instead.”

This leads to results resulting from these tests: “Agreement between nurse and Biobeat detection of these abnormalities was generally low, except for MBP < 60 mmHg, where Cohen’s kappa suggested fair agreement (0.30 [-0.04;0.64]). Severe hypotension (MBP <60 mmHg, the primary objective) was not more frequently detected with Biobeat than with nurse surveillance (P = 0.07, McNemar test) but this was the case when considering the threshold of 65 mmHg (P = 0.01). Biobeat also showed higher rates of discovery for PR ≥ 100 beats/min (P < 0.001), RR > 20 breaths/min (P < 0.001), SpO2 up to > 90% (P < 0.001), and temperature <36,8° or >38 °C (P < 0.001).”

Comment 10: What are the "1.1% serious issues" in sensor installation?

Response to comment 10: We modified the paragraph: “Sensor installation was straightforward in 76 cases. Minor issues occurred in 10 patients, and moderate issues occurred in three patients. The sensor was not recognized by the platform in one case which prevented its configuration in the PACU; the configuration was subsequently completed when the patient was in the surgery department.”

Comment 11: What data support "the sensor outperformed conventional monitoring in detecting vital sign abnormalities" in this study?

Response to comment 11: This is supporting by data presented in Table 3 and analysis using McNemar test: Biobeat identifies more patients with postoperative hypotension (<65 mmHg), tachycardia (≥100 beat/min), polypnea (>20 breaths/min), oxygen desaturation (SpO2 < 90%) and temperature (<36,8°ot >38°) than nurse. We have modified the first paragraph of the Discussion section as follows: “This study of the Biobeat wearable sensor in postoperative patients, conducted without human intervention, produced mixed results: significant data loss, few artifacts, and varied agreement with nurse measurements across parameters. However, the sensor allows for more frequent detection of several vital sign abnormalities than conventional nursing monitoring.”

We also emphasized that the population studied was "not very serious" in the chapter. “Ability to detect vital sign abnormalities relative to standard care” as follows: “This was not the case in our study, as shown by the low number of patients, or even their absence, when considering the occurrence of very severe hypotension (MAP < 60 mmHg), bradycardia (PR ≤ 40 beats/min), bradypnea (RR < 8 breaths/min), or hyperthermia (temperature > 39°C).”

Comment 12: - I am not sure what authors mean with "real-world" study. The clinical protocol refers to a clinical study (which is of course real). In most studies involving wearables, "real-world" means "into the wild" ie outside the clinical environment, where there is little to no control of what the person is doing during monitoring. Please, clarify.

Response to comment 12: This term was indeed used inappropriately. We would like to emphasize that the patients were hospitalized in a conventional unit with standard nursing supervision. Wearing the Biobeat patch did not alter their care in any way; the only specific element was that a

---

## [Decision Letter · Decision Letter 1]

23 Dec 2025

Dear Dr. Fischler,

Thank you for submitting your manuscript to PLOS ONE. After careful consideration, we feel that it has merit but does not fully meet PLOS ONE’s publication criteria as it currently stands. Therefore, we invite you to submit a revised version of the manuscript that addresses the points raised during the review process.

We look forward to receiving your revised manuscript.

Kind regards,

Agnese Sbrollini

Academic Editor

PLOS One

Journal Requirements:

Reviewers' comments:

Reviewer's Responses to Questions

**Comments to the Author**

Reviewer #1: All comments have been addressed

Reviewer #3: (No Response)

2. Is the manuscript technically sound, and do the data support the conclusions?

Reviewer #1: (No Response)

Reviewer #3: Yes

3. Has the statistical analysis been performed appropriately and rigorously?

Reviewer #1: (No Response)

Reviewer #3: Yes

4. Have the authors made all data underlying the findings in their manuscript fully available?

Reviewer #1: (No Response)

Reviewer #3: Yes

5. Is the manuscript presented in an intelligible fashion and written in standard English?

Reviewer #1: (No Response)

Reviewer #3: Yes

Reviewer #1: (No Response)

Reviewer #3: The current manuscript reports the performance of a remote monitoring device. The reviewer was not part of the previous round(s) of reviews. The writing is generally good. The reviewer has the following comments:

1) Line 63: Equipment can be cited in-text with the company name and location without its website.

2) Introduction: Before introducing the Biobeat sensor, please expand the review of existing products with similar functionalities. Currently, only lines 59-62 satisfy this requirement.

3) Lines 81-83: Published guidelines should be cited as formal references.

4) The study design should describe the basic settings in the main text. You cannot put everything in the appendices.

5) More basic information should be provided for patients.

6) Please provide all figures using commonly used formats, such as PDF. Figure S4 is not readable. Why is it not part of the normal figures in text? Figures 1-5 can be read correctly.

7) Line 102: All abbreviations should be spelled out when being used for the first time in the text (there are other similar incidents in the manuscript). Citations are required for PPG.

8) Line 124: Web sources need to be cited as a formal reference, not just an in-text link.

9) Line 129: Spell out “hours”

10) The subsection of “outcomes” is not necessary and can be incorporated into the results section. The subsection of “sample size” is not necessary either and can be incorporated into “patient population”.

11) “Statistical Analysis”: All named methods mentioned in this section should be briefly explained and cited.

12) Table 1: This is not part of the results. This should be part of the “patient population” information. See 5) above.

13) Figure 2: This should be part of “data collection”.

14) Figure 3 is missing?

15) “Biobeat Sensor Artifacts”: How did you define an artifact? There must be an objective definition. This should be defined in the Method section first.

16) “Biobeat Sensor Artifacts”: If the percentage is too small, increase the effective digits. It does not make sense to have a number, but the percentage is zero.

17) Figure 5 is missing?

18) The font of section headings is a little confusing. Please add section numbers (1.1, 1.2, etc.)

19) Some of the “discussion” should be moved to results, as the discussion section is further explanation/analysis of results. No new data should appear in the discussion.

.

Reviewer #1: No

Reviewer #3: No

You may also use PLOS’s free figure tool, NAAS, to help you prepare publication quality figures: https://journals.plos.org/plosone/s/figures#loc-tools-for-figure-preparation

---

## [Author Response · Author response to Decision Letter 2]

7 Jan 2026

Response to the Reviewer 3

General comment

The current manuscript reports the performance of a remote monitoring device. The reviewer was not part of the previous round(s) of reviews. The writing is generally good. The reviewer has the following comments.

Response to the general comment:

We thank the reviewer for their comments and have amended the manuscript accordingly. However, some comments relate to formatting and these sometimes conflict with the journal's recommendations. This is noted below.

Some phrasing has been revised.

Deleted elements in the modified text are in red and strikethrough; added/replacement text is highlighted in blue.

Comment 1) Line 63: Equipment can be cited in-text with the company name and location without its website.

Response to Comment 1): Modification has been made as requested: “The Biobeat system (Biobeat Technologies Ltd., Petah Tikva, Israel) …”

Comment 2) Introduction: Before introducing the Biobeat sensor, please expand the review of existing products with similar functionalities. Currently, only lines 59-62 satisfy this requirement.

Response to Comment 2): Introduction: We have modified the introduction to address your comment: “Since the early 2000s, the rise of broadband Internet, Wi Fi, and Bluetooth has transformed remote patient monitoring from occasional telemedicine experiments into structured systems capable of continuously measuring and transmitting patients’ clinical parameters such as blood pressure (BP), heart rate (HR), pulse rate (PR), respiratory rate (RR), peripheral oxygen saturation (SpO2), and temperature. Numerous wearable devices now enable the simultaneous capture of multiple vital signs and transmit them in real time to monitoring platforms or electronic medical records. A recent narrative review by Bignami et al. described nine clinically validated multiparametric monitoring devices [5]. They show that, although the dividing line remains debatable, some are primarily designed for in hospital use (CheckPoint Cardio CPC12S [6], Lifetouch [7], Portrait Mobile [8], Radius VSM [9], SensiumVitals [9], VitalPatch [10]) while others are aimed more at out of hospital settings, particularly prehospital environments and the home (BioButton [11], CardioWatch 287-2 [12], C-Med Alpha [13].”

Comment 3) Lines 81-83: Published guidelines should be cited as formal references.

Response to Comment 3): We believe that you are referring to the protocol and to the ethics committee agreement, which are on lines 81-83 of the submitted PDF. However, if these items are cited as formal references, the reader will no longer have access to them. That is why we prefer to keep the current format, but perhaps we have misunderstood your comment.

Comment 4) The study design should describe the basic settings in the main text. You cannot put everything in the appendices.

Response to Comment 4): In response to this comment, we have modified the text:

- we specified more precisely the inclusion criteria: « Adults (≥ 18 years of age), classified American Society of Anesthesiologists physical status 1 to 3, undergoing major surgery (digestive, gynecological, orthopedic, or urological surgery with expected durations of > 2 hours and a postoperative stay of ≥ 2 nights) were eligible regardless of their comorbidity.”

- we have retained a chapter entitled "Study procedure" which covers both monitored surveillance and nursing surveillance.

This approach provides readers with access to essential protocol elements without requiring consultation of the Appendix.

Comment 5) More basic information should be provided for patients.

Response to Comment 5): We have clarified this in the corrected version: “Written informed consent was obtained from all participants prior to enrollment, after the study protocol (including duration, methods, and outcome assessment) had been explained and the device presented.”

Comment 6) Please provide all figures using commonly used formats, such as PDF. Figure S4 is not readable. Why is it not part of the normal figures in text? Figures 1-5 can be read correctly.

Response to Comment 6): According to PLOS One formatting requirements, figures must be submitted in TIFF or EPS format only (https://journals.plos.org/plosone/s/figures#loc-file-format).

Regarding Figure S4: This figure is present in the list of submitted files and in the complete PDF (page 56/96) that was returned to us for confirmation of article submission acceptance. Figure S4, which illustrates the Biobeat patch, was originally Figure 1 in the initial submission but was moved to supplementary materials following a directive from the journal editorial board received on August 5, 2025: "Please note that you must upload a completed CONSORT flowchart as Figure 1 of your manuscript." We were therefore obliged to follow this requirement. Since this figure logically belonged in the device description section, the supplementary materials format was the only appropriate placement.

Comment 7) Line 102: All abbreviations should be spelled out when being used for the first time in the text (there are other similar incidents in the manuscript). Citations are required for PPG.

Response to Comment 7): We have verified that each abbreviation is spelled out in full when first mentioned.

The manuscript already included a reference to the use of PPG for cardiovascular assessment (lines 62-63) however, we have added a reference for general PPG applications. We have modified the text as follows: “Devices such as Biobeat, CardioWatch 287-2 [14] and Checkpoint Cardio’s CPC12S [6] also approximate blood pressure (BP) using photoplethysmography (PPG) [15], which is a widely adopted method for portable cardiovascular assessment [16].”

Comment 8) Line 124: Web sources need to be cited as a formal reference, not just an in-text link.

Response to Comment 8): Modified as requested. The platform name is now presented in the text, and the website is cited as a formal reference.

Comment 9) Spell out “hours”

Response to Comment 9): Modified as requested. We have applied this change consistently throughout the entire manuscript.

Comment 10) The subsection of “outcomes” is not necessary and can be incorporated into the results section. The subsection of “sample size” is not necessary either and can be incorporated into “patient population”.

Response to Comment 10): The STROBE Statement explicitly requires "Clear definition of all outcomes" and "Study size: Explanation of how the study size was determined." This is why our manuscript retains these subsections in accordance with reporting guidelines.

Comment 11) “Statistical Analysis”: All named methods mentioned in this section should be briefly explained and cited.

Response to Comment 11): This paragraph has been revised as requested: “For continuous paired measurements, nurse-recorded values and the closest corresponding Biobeat values were compared using the Bland-Altman method for repeated measures, which assesses agreement between two quantitative methods by plotting differences against means and estimating within- and between-subject variance–adjusted limits of agreement [32]. Comparisons were not performed if the interval between these measurements was greater than 5 minutes. These paired measurements were also evaluated for clinical accuracy using Clarke Error Grid analysis, originally developed for self-monitoring of blood glucose [33]. A standard scatter plot of the sensor and nurse recordings was generated and divided into five zones (A–E). Zones A and B indicate acceptable agreement, and zones C–E indicate increased discrepancies. Clarke Error Grid zones were defined according to the authors’ clinical judgment and criteria from prior validation studies [9,34–36]. For binary paired data, agreement between nurse and Biobeat classifications was evaluated using Cohen’s kappa coefficient, which quantifies agreement beyond chance but is sensitive to the distribution of positive and negative categories [37]. To determine whether discordant pairs favored one method (whether nurse or patch produces significantly more positive or negative results than the other), McNemar’s test with continuity correction was applied [38]. When the total number of discordant pairs was <10, the exact McNemar test was used instead.”

Comment 12) Table 1: This is not part of the results. This should be part of the “patient population” information. See 5) above.

Response to Comment 12): We have incorporated key demographic and clinical information into the population description: “Adults (≥ 18 years of age), classified American Society of Anesthesiologists physical status 1 to 3, undergoing major surgery (digestive, gynecological, orthopedic, or urological surgery with expected durations of > 2 hours and a postoperative stay of ≥ 2 nights) were eligible regardless of their comorbidity.” However, we have retained a summary table of principal patient characteristics, as is standard practice in clinical research reporting.

Comment 13) Figure 2: This should be part of “data collection”.

Response to Comment 13): Please see our response to your comment 6.

We were required to follow the directive received from PLOS on August 5, 2025: "Please note that you must upload a completed CONSORT flowchart as Figure 1 of your manuscript."

This requirement necessitated the repositioning of all subsequent figures after Figure 1.

Comment 14) Figure 3 is missing?

Response to Comment 14): Figure 3 was present in the complete PDF that was returned to us for confirmation of article submission acceptance (page 50/96). We apologize for any confusion caused by the apparent reversal between Figures 3 and 4 (in the PDF).

Figure 3 is page 48 of the present version.

Comment 15) “Biobeat Sensor Artifacts”: How did you define an artifact? There must be an objective definition. This should be defined in the Method section first.

Response to Comment 15): We provided objective definitions in the first sentence of the Statistical analysis subsection. Recognizing this was suboptimal placement, we have relocated this text to the end of the "Data Collection" section for clarity.

Comment 16) “Biobeat Sensor Artifacts”: If the percentage is too small, increase the effective digits. It does not make sense to have a number, but the percentage is zero.

Response to Comment 16): We have revised the percentages to be expressed per 1,000. Thank you for pointing out this reporting issue.

Comment 17) Figure 5 is missing?

Response to Comment 17): Figure 5 was present in the complete PDF that was returned to us for confirmation of article submission acceptance (page 50/96). We again apologize for any confusion caused by the reversal between Figures 3 and 4.

Figure 5 is page 50 of the present version.

Comment 18) The font of section headings is a little confusing. Please add section numbers (1.1, 1.2, etc.)

Response to Comment 18): We have followed the formatting recommendations specified by PLOS One in their official guidelines (PLOSOne_formatting_sample_main_body.pdf).

Comment 19) Some of the “discussion” should be moved to results, as the discussion section is further explanation/analysis of results. No new data should appear in the discussion.

Response to Comment 19): While we respectfully acknowledge your concern, we note that our discussion section includes recalled results primarily to compare them with findings in the literature, which is standard practice. However, we have removed the sentence "A similar value (73.7%) was found in our study" because this metric was not analyzed using the same comparative methodology for other variables.

---

## [Decision Letter · Decision Letter 2]

19 Jan 2026

Dear Dr. Fischler,

Thank you for submitting your manuscript to PLOS ONE. After careful consideration, we feel that it has merit but does not fully meet PLOS ONE’s publication criteria as it currently stands. Therefore, we invite you to submit a revised version of the manuscript that addresses the points raised during the review process.

We look forward to receiving your revised manuscript.

Kind regards,

Agnese Sbrollini

Academic Editor

PLOS One

Journal Requirements:

Reviewers' comments:

Reviewer's Responses to Questions

**Comments to the Author**

Reviewer #1: All comments have been addressed

Reviewer #4: All comments have been addressed

Reviewer #5: All comments have been addressed

2. Is the manuscript technically sound, and do the data support the conclusions?

Reviewer #1: (No Response)

Reviewer #4: Partly

Reviewer #5: Yes

3. Has the statistical analysis been performed appropriately and rigorously?

Reviewer #1: (No Response)

Reviewer #4: Yes

Reviewer #5: Yes

4. Have the authors made all data underlying the findings in their manuscript fully available?

Reviewer #1: (No Response)

Reviewer #4: Yes

Reviewer #5: Yes

5. Is the manuscript presented in an intelligible fashion and written in standard English?

Reviewer #1: (No Response)

Reviewer #4: No

Reviewer #5: Yes

Reviewer #1: (No Response)

Reviewer #4: * 16% of the SpO₂ data were missing, which limits accurate evaluation of oxygen saturation trends. This gap may underestimate the device’s performance in detecting hypoxemia.

* The median recording period of about 64 hours limits insights into long-term performance, connectivity stability, and patient adherence.

Reviewer #5: The revised manuscript entitled “Continuous remote monitoring of postoperative vital signs with the Biobeat patch: A multicenter prospective observational study” represents a strong and carefully executed contribution to the literature on postoperative wireless monitoring. I commend the authors for their comprehensive revisions following the previous review round.

Strengths

The study design is transparent, well‑structured, and fully aligned with STROBE reporting principles.

The description of the monitoring protocol, device placement, artefact identification, calibration, and data‑handling procedures is clear and accurate.

Statistical analyses are rigorous and now fully explained, including appropriate use of Bland–Altman, Clarke Error Grid, Cohen’s kappa, and McNemar’s testing. [Notes | Txt]

The availability of all underlying data in Dryad meets PLOS requirements.

The discussion presents a balanced assessment of strengths, limitations, and implications for clinical practice.

Areas Improved in This Revision

First‑mention definitions of all abbreviations and clearer referencing of PPG methodology.

Enhanced introduction summarising comparable wearable systems.

Improved description of study procedures and inclusion criteria, now appropriately integrated into the main text.

Better clarity regarding artefact definitions and their placement within the Data Collection section.

Minor Suggestions (Non‑blocking)

A brief tightening of certain sentences in the Discussion could further enhance clarity, although the current version is acceptable.

Consider removing any residual repetition regarding consent and data transmission procedures.

Overall Assessment

This is a methodologically sound and clinically relevant study with implications for postoperative surveillance and the practical challenges of continuous wireless monitoring. The authors have addressed all major concerns raised earlier, and the manuscript is suitable for publication pending minor editorial adjustments.

.

Reviewer #1: No

Reviewer #4: No

Reviewer #5: No

---

## [Author Response · Author response to Decision Letter 3]

6 Feb 2026

We sincerely thank the Editorial team at PLOS ONE and the reviewers for their thoughtful evaluation of our manuscript. This response details our point-by-point replies to the reviewer’s comments and summarizes the modifications made to the manuscript. Deleted sentences are in red and crossed out in the marked text, while added sentences are in blue.

In addition, some words and sentences have been modified to make the text flow better; they are reported in blue in the marked file.

Reviewer #4

Comment 1: 16% of the SpO₂ data were missing, which limits accurate evaluation of oxygen saturation trends. This gap may underestimate the device’s performance in detecting hypoxemia.

Response to Comment 1: We appreciate this important observation and recognize that data completeness is a key metric of technical performance. We have substantially revised the manuscript to clarify the nature, quantification, and explanation of SpO2 data loss.

First, we modified the chapter “Biobeat Monitoring System Data collection” as follows: "Of 64,088 expected simultaneous multi parameter physiological measurements at 5 minute intervals, 28,090 (43.8%) were missing, while 29,514 contained complete data for all variables. Among the 6,484 incomplete serial measurements, 89.6% lacked SpO2, 11.1% lacked MBP, and 8.9% lacked RR. PR and temperature were missing in only 7 and 3 instances, respectively. Overall, SpO2 was the most frequently absent variable, occurring in 16.1% of all obtained serial measurements, either in isolation or alongside other missing parameters. Fig 3 illustrates an example of complete monitoring and incomplete data with periodic gaps."

Second, we have completed the chapter on "Data loss" in the Discussion section as follows: “Finally, one particular point concerns SpO2, which emerged as the parameter most frequently missing when data transmission has been satisfactory. SpO2 is intrinsically difficult to measure accurately because the arterial pulsatile component of PPG represents only a small fraction of the overall optical signal and is easily contaminated by noise. Motion artifacts, low peripheral perfusion, vasoconstriction, ambient light, and sensor misplacement can all distort the PPG waveform and lead to erroneous SpO₂ estimates which are rejected by the device [41]”.

Comment 2: The median recording period of about 64 hours limits insights into long-term performance, connectivity stability, and patient adherence.

Response to Comment 2: We agree that the 64-hour median monitoring period represents a limitation and appreciate the reviewer's point about long-term performance. However, we believe this duration is appropriate for the study's objectives and is comparable to the existing literature. We have substantially revised the manuscript to contextualize this timing. This is discussed in the chapter Strengths and limitations as follows in the revised manuscript: “Third, we included patients with an expected postoperative hospital stay of ≥ 2 nights. This criterion was based on evidence that the majority of postoperative complications (approximately 60%) occur within 1–3 days after surgery, as demonstrated in a cohort study of patients undergoing intra-abdominal operations [55]. However, future prospective studies should evaluate extended monitoring in patients requiring prolonged postoperative supervision (e.g., those undergoing complex surgeries or with severe comorbidities), with monitoring extending at least through the first week post-discharge.”

Reviewer #5

Comment 1: First mention definitions of all abbreviations and clearer referencing of PPG methodology.

Response to Comment 1:

We have verified that all abbreviations are defined when they first appear and we gave a brief overview of PPG technology: “Blood pressure can be measured using photoplethysmography (PPG) which is a non-invasive optical technique that detects changes in microvascular blood volume through analysis of light absorption variation. The direct current (DC) component represents baseline tissue absorbance, whereas the alternating current (AC) component reflects cardiac-synchronous arterial blood volume changes. This enables reliable extraction of SpO2, PR, and RR through validated algorithms, with the perfusion index (AC/DC ratio) serving as a clinical indicator of peripheral perfusion status. Waveform morphology analysis permits non-invasive assessment of arterial stiffness through pulse wave features [13]. Wearable devices such as Biobeat [14] and CardioWatch 287-2 [15] have demonstrated blood pressure estimation capabilities through PPG analysis in controlled, calibrated settings, though pulse transit time-based BP estimation encounters substantial technical limitations, including pre-ejection period contamination and insufficient signal information content, with long-term ambulatory accuracy remaining inadequately characterized. The Checkpoint Cardio CPC12S attempted blood pressure measurement through PPG but demonstrated unacceptably low accuracy in validation testing, exemplifying the current limitations of cuffless BP estimation from optical signals [5].”

Comment 2: Enhanced introduction summarising comparable wearable systems.

Response to Comment 2:

We have modified this paragraph: “Numerous wearable devices can capture multiple vital signs simultaneously and transmit data in real time to monitoring platforms or electronic medical records. In a recent narrative review, Bignami and colleagues identified nine clinically validated multiparametric monitoring systems. Although distinctions are not absolute, several devices are designed primarily for in hospital use (CheckPoint Cardio’s CPC12S, Lifetouch, Portrait Mobile, Radius VSM, SensiumVitals, and VitalPatch [5-9]), whereas others are oriented toward out of hospital applications, particularly prehospital and home monitoring (BioButton, CardioWatch 287 2, and C Med Alpha [10-12]).”

Comment 3: Improved description of study procedures and inclusion criteria, now appropriately integrated into the main text. Better clarity regarding artefact definitions and their placement within the Data Collection section.

Response:

We have modified the Data Collection section and the Outcome section to respond to this comment.

- Revised Data Collection section: “Each patient was assigned a unique identifier at sensor installation. Demographic data and nurse measurements with corresponding timestamps were extracted from hospital medical records. Although Biobeat real-time data display was available, it was not utilized during this study. Biobeat data were retrieved post-study from the Instamed company [24]. We aimed to obtain synchronous measurements of vital signs (PR, MBP, RR, SpO2, and temperature) at 5 minute intervals. Artifact frequency (secondary outcome indicating technical performance) was particularly examined, with objective definitions based on previously published criteria. Artifacts were defined as values exceeding 50% deviation from the prior reading (unless followed by a ±25% recovery value) or falling outside physiological ranges (e.g., PR <5 or >250 bpm; systolic BP <20 or >300 mmHg; SpO2 change ≥8% between consecutive readings) [25-28]. After numerical analysis, four independent experts reviewed the tracings to identify additional artifacts not captured by the automated criteria.”

- Revised Outcome section: “The primary outcome was the proportion of patients with major hemodynamic abnormalities defined as mean blood pressure (MBP) <60 mmHg, detected by the Biobeat patch. The secondary outcome was the proportion of patients with vital sign abnormalities at various predefined thresholds as detected by either nurse monitoring or the Biobeat device. Other secondary outcomes include data loss and artifact frequency (technical performance), incidence of postoperative complications, assessment of sensor installation ease, and evaluation of patient acceptability and post-removal skin reactions.”

Minor Suggestions: A brief tightening of certain sentences in the Discussion could further enhance clarity, although the current version is acceptable.

Consider removing any residual repetition regarding consent and data transmission procedures.

Response to minor suggestions:

We reviewed the Discussion and tried to make the text clearer and removed repetition.

---

## [Decision Letter · Decision Letter 3]

24 Feb 2026

Continuous remote monitoring of postoperative vital signs with the Biobeatpatch: A multicenter prospective observational study

PONE-D-25-42386R3

Dear Dr. Fischler,

We’re pleased to inform you that your manuscript has been judged scientifically suitable for publication and will be formally accepted for publication once it meets all outstanding technical requirements.

Kind regards,

Agnese Sbrollini

Academic Editor

PLOS One

Additional Editor Comments (optional):

Reviewers' comments:

Reviewer's Responses to Questions

**Comments to the Author**

Reviewer #1: All comments have been addressed

Reviewer #4: All comments have been addressed

2. Is the manuscript technically sound, and do the data support the conclusions?

Reviewer #1: (No Response)

Reviewer #4: Yes

3. Has the statistical analysis been performed appropriately and rigorously?

Reviewer #1: (No Response)

Reviewer #4: Yes

4. Have the authors made all data underlying the findings in their manuscript fully available?

Reviewer #1: (No Response)

Reviewer #4: Yes

5. Is the manuscript presented in an intelligible fashion and written in standard English?

Reviewer #1: (No Response)

Reviewer #4: Yes

Reviewer #1: (No Response)

Reviewer #4: Accepted — all reviewer comments have been fully addressed. I would like to thank the authors for their efforts

.

Reviewer #1: No

Reviewer #4: No

---

## [Editor Report · Acceptance letter]

PONE-D-25-42386R3

PLOS One

Dear Dr. Fischler,

I'm pleased to inform you that your manuscript has been deemed suitable for publication in PLOS One. Congratulations! Your manuscript is now being handed over to our production team.

Kind regards,

on behalf of

Dr. Agnese Sbrollini

Academic Editor

PLOS One